# Factors Associated with Viral Load Suppression and Indicators of Stigma among People Living with HIV in Dar es Salaam Tertiary Hospitals, Tanzania

**Mary Spicar Kilapilo** [1], **Idda Hubert Mosha** [2], **George Msema Bwire** [1], **Godfrey Leonard Sambayi** [3]
**Raphael Zozimus Sangeda** [1,*] and **Japhet Killewo** [4]

1  Department of Pharmaceutical Microbiology, Muhimbili University of Health and Allied Sciences,
   Dar es Salaam P.O. Box 65013, Tanzania; kilapilomary468@gmail.com (M.S.K.)
2  Department of Behavioural Sciences, Muhimbili University of Health and Allied Sciences,
   Dar es Salaam P.O. Box 65001, Tanzania
3  Department of Pharmacognosy, Muhimbili University of Health and Allied Sciences,
   Dar es Salaam P.O. Box 65013, Tanzania
4  Department of Biostatistics and Epidemiology, Muhimbili University of Health and Allied Sciences,
   Dar es Salaam P.O. Box 65001, Tanzania
*  Correspondence: sangeda@gmail.com

**Abstract:** The perception of stigma can contribute to virological failure among people living with HIV (PLHIV). This study was conducted to find out how stigmatization and self-stigma affect the ability of people living with HIV (PLHIV) in Dar es Salaam, Tanzania, to keep their viral load down. This was a hospital-based cross-sectional study conducted in Temeke Regional Referral Hospital (RRH) and Amana RRH at the Care and Treatment Clinic (CTC) between July and August 2020 using a structured questionnaire with open- and close-ended questions. Multivariate logistic regression analysis was used to examine the factors of viral load suppression. The Chi-square test was used to compare the factors of stigmatization and viral load suppression. Altogether, 406 PLHIV participated, with the most being female respondents, 298 (73.2%). The majority (50%) were aged between 25 and 44 years, whereas 171 (42.5%) respondents were married. Most of the participants, 382 (94.6%), were on a dolutegravir-based regimen, with the majority, 215 (52.8%), having a refill interval of three months. Most respondents, 379 (93.1%), disclosed their status. Most participants, 355 (87.4%), preferred having a separate HIV clinic, while 130 (32.1%) participants were not ready to be attended by the health care workers (HCWs) familiar to them. Male patients were 60% less likely to suppress their viral load as compared to female patients (adjusted odds ratio [aOR]: 0.4, 95% confidence interval [95%]: 0.19–0.77, *p*-value = 0.007). The refill interval was significantly associated with viral load suppression. For example, patients with a one-month refill interval had odds of 0.01 (95% CI: 0.003–0.42, *p*-value = 0.0001) compared to six-month refill intervals. Stigmatization elements appeared to influence viral load suppression among PLHIV in the Dar es Salaam area, significantly predicting viral load outcomes when gender and time between refills were considered.

**Keywords:** self-discrimination; stigmatization; Dolutegravil; Dar es Salaam; HIV; viral load suppression; people living with HIV; Tanzania

## 1. Introduction

Viral load suppression indicates disease progression, wellness, antiretroviral therapy (ART) effectiveness, and reduced risk of HIV drug resistance (HIVDR). Virological failure, on the other hand, indicates the inverse and is associated with HIVDR [1]. Stigmatization was found to have a significant link with non-adherence after the characterization of the factors contributing to non-adherence [2–4].

People living with HIV (PLHIV) continue to face stigma and discrimination from their families and communities. According to research, HIV-related stigma and self-stigmatization delay the disclosure of HIV serostatus, which may be a barrier to HIV counseling, staying in care and treatment, and receiving and using ART [5–7]. AIDS-related stigma and discrimination prevent millions of PLHIV from accessing and benefiting from effective prevention and treatment services [6,8].

Self-stigmatization among PLHIV is one outcome of fear of community stigma. Consequently, PLHIV may feel pressured to hide their serostatus and, in many cases, to do things that put them at risk [9]. Therefore, this study examined how stigmatization and self-discrimination affect viral load suppression in PLHIV who attend care and treatment clinics at Amana and Temeke RRHs in Dar es Salaam, Tanzania.

## 2. Materials and Methods

### 2.1. Study Design and Area and Period of the Study

Between July and August 2020, a cross-sectional study was conducted at Temeke Regional Referral Hospital (RRH) and Amana RRH at the Care and Treatment Clinic (CTC) in Dar es Salaam, Tanzania. Dar es Salaam city was selected to represent the Tanzanian mainland regions. Dar es Salaam is Tanzania's largest city, economic hub, and former capital, with a population of approximately five million (about 10% of the country's total population) [10].

### 2.2. Study Population and Sampling

This study included 406 PLHIV over 18 years who attended Amana and Temeke RRHs and were on ART for at least six months. The sample size was calculated using the cross-sectional formula [11] and the assumed prevalence from the previous study [12]. Amana RRH CTC provides medical care to 8800 PLHIV, while Temeke RRH CTC provides medical care to 6500 PLHIV. During the recruitment of research participants, a systematic approach was utilized. This was performed by obtaining a sampling interval "n" as the total number of patients on CTC divided by 200 and by the number of study days. Then, a patient was recruited after every "n" interval by selecting the nth patient.

### 2.3. Data Collection Tool

A structured questionnaire with open- and close-ended questions was prepared following an intensive literature review on a topic related to stigmatization, adherence, and viral load suppression [11–13]. The English questionnaire was designed and translated into Kiswahili (the local language). The Kiswahili questionnaire was uploaded on REDCap (Research Electronic Data Capture). REDCap is an electronic data capture tool hosted at Muhimbili University of Health and Allied Sciences (MUHAS) [14,15]. The questionnaire was tested on a pilot population of 30 patients (15 from each CTC). The updated Kiswahili questionnaire was then used to obtain information about patients' demographics, pharmacy refills, and adherence to antiretroviral therapy (ART). Tablets were used to collect data, which was then translated into English and coded in the REDcap. The viral load measurements were obtained from CTC medical records. REDCap data were downloaded, cleaned in Excel (to remove incomplete forms) and then exported to the Statistical Package for Social Sciences (SPSS software version 25, Chicago Inc., Chicago, IL, USA) for analysis.

### 2.4. Data Management and Analysis

Descriptive statistics were summarized using frequencies and percentages. Stigmatization was estimated using the HIV stigma toolkit described elsewhere [16]. The association between categorical variables such as gender, refill interval, regimen used, HCWs' preference, and viral load suppression was evaluated using a Chi-square test. Viral load suppression and high viral load were defined as viral load counts below 1000 copies/mL and above 1000 copies/mL, respectively [17]. Factors associated with viral load suppression were controlled using a binary regression model. Factors with a *p*-value less than 0.2 in

univariate regression were qualified for multivariate logistic regression [18]. A *p*-value of less than 0.05 was considered statistically significant.

## 3. Results

*3.1. Socio-Demographic Characteristics of Participants*

This study included 406 people living with HIV, including 205 from Amana RRH and 201 from Temeke RRH. The majority of respondents were between the ages of 25 and 44 (50%) and were predominantly female (73.2%). The vast majority, 151 (38.2%), of those investigated had been on ART for 7 to 12 years. Furthermore, the majority of study participants, 171 (42.5%), were married, and the majority of participants, 382 (94.6%), were on a dolutegravir-based regimen. The vast majority, 215 (52.8%), had a three-month refill interval (Table 1).

**Table 1.** Participants' socio-demographic characteristics.

| Variable | Category | Frequency N (%) |
| --- | --- | --- |
| Gender (n = 406) | Female | 298 (73.2) |
| | Male | 108 (26.5) |
| Age (n = 404) | 18–24 | 33 (8.2) |
| | 25–44 | 202 (50.0) |
| | 45–54 | 108 (26.7) |
| | 55–64 | 48 (11.9) |
| | ≥65 | 13 (3.2) |
| Hospital facility (n = 406) | Amana RRH | 205 (50.4) |
| | Temeke RRH | 201 (49.4) |
| Start of ART (years) (n = 395) | <2 | 80 (20.3) |
| | 3–6 | 103 (26.1) |
| | 7–12 | 151 (38.2) |
| | ≥12 | 61 (15.4) |
| Marital status (n = 402) | Married | 171 (42.0) |
| | Single | 106 (26.0) |
| | Divorced | 79 (19.4) |
| | Widowed | 46 (11.3) |
| On the dolutegravir-based regimen (n = 404) | Yes | 382 (94.6) |
| | No | 22 (5.4) |
| Refill interval (months) (n = 406) | 1 | 83 (20.4) |
| | 2 | 7 (1.7) |
| | 3 | 215 (52.8) |
| | 6 | 101 (24.8) |
| Cost for a single hospital visit (TZS) (n = 405) | <2000 | 283 (69.9) |
| | ≥2000 | 87 (21.4) |
| | No cost | 35 (8.6) |
| Waiting time at CTC (hours) (n = 405) | <0.5 | 51 (12.6) |
| | 0.5–1 | 137 (33.8) |
| | 1–2 | 128 (31.6) |
| | >2 | 89 (22) |
| Travelling time from home to CTC (hours) (n = 406) | <0.5 | 105 (25.9) |
| | 0.5–1 | 188 (46.3) |
| | 1–2 | 91 (22.4) |
| | >2 | 22 (5.3) |

Key: RRH: regional referral hospital.

### 3.2. Status Disclosure among People Living with HIV

Most of the respondents, 378 (93.1%), disclosed their HIV status. The majority, 352 (93.6%) of those disclosing their HIV status, were consequently encouraged to take medication. Most of the participants, 246 (61.2%), reported concealing their HIV status to themselves, while 212 (52.7%) reported not having a reminder person or tool to assist them in taking their medication on time. When asked about the reminder person or the reminder tool, the majority of the participants, 52.7%, reported a lack of reminders. In contrast, the remaining participants, 24.4%, said their partners remind them. In comparison, 19.7% are reminded by their family members, and 6.7% reported using a phone and other devices to remind them to take medication (Table 2).

**Table 2.** Status disclosure among people living with HIV.

| Variable | Categories | Frequency (%) |
|---|---|---|
| Disclosure of HIV status | Yes | 378 (93.1) |
| (n = 406) | No | 28 (6.9) |
| | They encouraged me to take medication | 352 (93.6) |
| Response after disclosing the status. | They supported me financially and socially | 111(29.5) |
| (n = 490) * | They left me stigmatized | 15 (4.0) |
| | They left me to deal with it alone | 12 (3.2) |
| Preference to keep secret their HIV status. | Prefer to keep a secret | 246 (61.2) |
| (n = 402) | Do not prefer to keep a secret | 156 (38.8) |
| | No one reminds me | 212 (52.7) |
| Who is the reminder person to assist in taking medication | Yes, I am reminded by someone | 167 (41.3) |
| (n = 406) | The phone alarm reminds me | 27 (6.0) |

Key: * multiple responses were given.

### 3.3. Self-Discrimination and Stigmatization

Of all participants, 355 (87.4%) preferred having a separate HIV clinic for HIV patients. When asked for a reason, most participants, 309 (87.0%), reported that a separate HIV CTC is a free space for HIV patients. Most participants, 275 (67.9%), agreed on having HIV hospital services if the HCW was a familiar person from their neighborhood (Table 3).

### 3.4. Proportions of Patients with Viral Load Suppression per Social Demographic Characteristic

When the factors that affect viral load suppression were evaluated, it was found that viral suppression was higher in female patients than in male patients (*p*-value = 0.036). Those on the dolutegravir-based regimen also suppressed viral load more than those on other regimens (*p*-value = 0.050). Participants with a refill interval of six months had a higher proportion of viral suppression than those with three months and other refill intervals, with a *p*-value of less than 0.0001. Participants who would continue with service if the HCW is a familiar person from their neighborhood had more viral suppression than those who would prefer otherwise, with a *p*-value = 0.004 (Table 4).

### 3.5. Multivariate Logistic Regression Analysis of Factors Associated with Viral Load Suppression

Upon multivariate analysis, male patients were 60% less likely to have viral load suppression compared to female patients (adjusted odds ratio [aOR]: 0.4, 95% confidence interval [95%]: 0.19–0.77, *p*-value = 0.007). The refill interval was significantly associated with viral load suppression. For example, patients with one-month refill intervals had odds of 0.01 (95% CI: 0.003–0.42, *p*-value = 0.0001) compared to those with six-month refill intervals (Table 5).

**Table 3.** Self-discrimination and stigmatization among 406 PLHIV.

| Variable | Categories | Frequency (%) |
|---|---|---|
| Preferred CTC location (n = 406) | Having separated HIV clinic | 355 (87.4) |
| | Inclusive HIV clinic | 26 (6.4) |
| | Anyhow is better | 18 (4.4) |
| | None of the above | 7 (1.7) |
| Why prefer separated HIV CTC * (n = 406) | It is a free space for HIV patients | 309 (87.0) |
| | I am comfortable seeing others with HIV, knowing that I am not alone | 99 (27.9) |
| | I just like how things are | 35 (9.9) |
| | So as to avoid the long waiting queue at the inclusive clinic | 28 (7.9) |
| Would you have your HIV hospital services if the health care worker is a person you know from your neighbourhood? | Yes | 275 (67.8) |
| | No | 131 (32.2) |
| Why not receive the hospital service from familiar HCWs from the neighbourhood * (n = 131) | Because of the stigma that I will receive | 56 (43.4) |
| | Because of fear that HCW will disclose my health status | 53 (41.1) |
| | Because I do not prefer people who know me to know my health status | 30 (23.3) |
| Which container is used to store the medication? (n = 406) * | Inside its container | 320 (78.8) |
| | Anywhere but not inside its container | 36 (8.9) |
| | Inside my handbag | 34 (8.4) |
| | Inside plastic bag | 30 (7.4) |
| | Inside an envelope | 15 (3.7) |
| | Inside the pockets of my clothes | 10 (2.5) |
| To avoid the noise made by the containers, what do you add to the container (n = 329 **) | I am not bothered by the sound; I do not do anything | 227 (73.7) |
| | I add a clean Handkerchief inside the container | 31 (10.1) |
| | I add leaflet paper inside the containers | 27 (8.8) |
| | I add clean cotton inside the container | 18 (5.8) |
| | I only take my medication when I am at home. I do not move with them | 16 (5.2) |
| | I do not move with medication inside the container. I keep them in the hospital plastic bags | 10 (3.2) |
| Where do you store antiretroviral medication at home? (n = 406) * | On top of the cupboard | 123 (30.3) |
| | Inside clothes cupboard | 118 (29.1) |
| | Under the bed | 83 (20.4) |
| | Inside backpack | 35 (8.6) |
| | In between clothes in the suitcase | 29 (7.1) |
| | Inside dishes cupboard | 19 (4.7) |
| | Inside the refrigerator | 7 (1.7) |
| | Near the window | 6 (1.5) |
| | Inside a drawer | 5 (1.2) |
| | In the car | 3 (0.7) |

Key: * multiple responses were given, ** missing data due to lack of responses.

**Table 4.** Proportions demographic characteristics categorized by viral load suppression using the Chi-square test among 406 PLHIV.

| Variable | Category | Total | Viral Load Suppressed n (%) | High Viral Load n (%) | *p*-Value |
|---|---|---|---|---|---|
| Gender (n = 385 **) | Female | 282 | 224 (79.4) | 58 (20.6) | **0.036** |
| | Male | 103 | 72 (69.9) | 31 (30.1) | |
| Regimen (n = 383 **) | DBR | 362 | 283 (78.2) | 79 (21.8) | **0.050** |
| | Another regimen | 21 | 11 (52.4) | 10 (47.6) | |
| Refill Interval (n = 385 **) | Six months | 99 | 95 (96) | 4 (4.0) | **<0.0001** |
| | Three months | 210 | 176 (83.3) | 34 (16.3) | |
| | One month | 69 | 23 (33.3) | 46 (66.7) | |
| | Two months | 7 | 2 (28.6) | 5 (71.4) | |
| HCW (n = 384 **) | They would continue with service if the HCW is a familiar person from their neighborhood | 255 | 207 (81.2) | 48 (18.8) | **0.004** |
| | They would NOT continue with hospital service if the HCW is a familiar person from their neighborhood | 129 | 88 (68.2) | 41(31.8) | |

Key: Bold *p*-values were statistically significant by Chi-square, HCWs: health care workers, DBR: dolutegravir-based regimen, ** missing data due to lack of viral load measurement.

**Table 5.** Multivariate logistics regression of factors associated with viral load suppression.

| Variable | Category | aOR (95%CI) | *p*-Value |
|---|---|---|---|
| Gender | Male | 0.4 (0.19–0.77) | **0.007** |
| | Female | Reference | **<0.0001** |
| Refill interval | One month | 0.01 (0.003–0.42) | **<0.0001** |
| | Two months | 0.01 (0.00–0.088) | **<0.0001** |
| | Three months | 0.2 (0.054–0.566) | **0.004** |
| | Six months | Reference | |
| Adherence | Adherent | 1.2 (0.552–2.408) | 0.705 |
| | Non-adherent | Reference | |
| Name of facility | Temeke RRH | 1.5 (0.735–3.02) | 0.269 |
| | Amana RRH | Reference | |
| Regimen used | DBR | 1.1 (0.125–9.384) | 0.942 |
| | Non-DBR | Reference | |

Key: Bold *p*-values were statistically significant, aOR: adjusted odds ratio, **95% CI**: 95% confidence interval, **RRH**: regional referral hospital, DBR: dolutegravir-based regimen.

## 4. Discussion

The study was conducted one year after the dolutegravir-based regimen was made the first-line treatment of choice for adults with HIV in Tanzania [17,19].

Our study aimed to investigate the socio-demographic characteristics, status disclosure, self-discrimination, and stigmatization among people living with HIV in Tanzania. The study included 406 participants living with HIV, and the majority were female and between the ages of 25 and 44. Most participants had been on ART for 7 to 12 years and were currently on a dolutegravir-based regimen.

Most participants (87.4%) in this study preferred a separate HIV clinic for HIV patients. The participants stated that a separate clinic would protect their privacy by concealing their daily medication from the surrounding community of PLHIV. It was also found that PLHIV frequently swapped around their containers of medications. Most PLHIV said they tucked their pills into plastic bags, envelopes, or purses. As a result, the ART's effectiveness decreases and there is a higher chance that the virus's replication will go

unchecked. However, we found no evidence that this factor played a significant role in the suppression of viral load in our study participants. This finding highlights the importance of creating safe spaces for people living with HIV, where they can receive quality healthcare services without fear of discrimination and stigmatization.

The majority of the participants (52.7%) reported a lack of a reminder person or tool to assist them in taking their medication on time, which could also contribute to suboptimal ART adherence. Using a reminder tool, such as a phone alarm, could be beneficial in improving ART adherence and reducing self-discrimination and stigmatization among people living with HIV [11].

Most study participants (30.3%) said they conceal their medication in a cupboard at home. This concurs with the findings of a study conducted in South Africa by Flynn and colleagues [20], in which most respondents said they kept their medication in either a refrigerator or a regular cupboard.

Additionally, our results showed that 78.2% of people treated with dolutegravir-based regimens had their viral load reduced to undetectable levels. This proportion is lower compared to a 2018 study conducted in Uganda, in which 94% of PLHIV on dolutegravir achieved viral load suppression [8]. This result also fell short of the national average for viral load suppression in Tanzania, where 87.0% of adults on ART reported doing so [17]. There's a chance that this result only applies to the first year after the dolutegravir regimen was implemented in Tanzania [19].

The female gender was significantly associated with better viral suppression. This finding contradicts the findings of other studies on the causes of undetected viral loads conducted in Uganda between 2014 and 2015 [21]. It is noteworthy that in Tanzania, the population of PLHIV consists mainly of females, accounting for more than two-thirds of the total population [11,22,23]. The proportion of females achieving viral suppression (79.4%) was higher compared to males (69.9%). However, this 9.5% difference did not meet the national average for viral load suppression in Tanzania, which was observed to be between 5% and 8% between males and females on ART in the 25–34 and 35–49 age groups, according to the 2016–2017 national impact survey, in ages 25–34 and ages 35–49, with a greater percentage of viral load suppression seen in females in both these age groups [24]. In the Tanzania National survey, a gender gap was also observed for each of the three 90s targets of UNAIDS, where women had a greater proportion compared to men on diagnosed (64.9% versus 52.2%), on treatment (95.3% versus 89.6%), and with suppressed viral loads (88.6% versus 83.2%) [24].

The Tanzania National Treatment Guideline recommends that PLHIV, who consistently maintain viral load suppression, be given a longer refill time interval [17], which may explain the significant association between viral load suppression and refill time interval found in this study. A similar strategy has been documented in Uganda [25].

When asked whether they would be willing to receive HIV hospital services from an HCW who is someone they know from their neighborhood, 67.9% of participants said they would be ready, while 32.1% would not. Concerns about being stigmatized and having their health status revealed were cited by 43.4% of PLHIV who were unwilling to be served by a familiar HCW, while the disclosure of their health status was cited by 41.1%. According to similar findings, HCWs' perpetuation of stigma may have a more significant impact on PLHIV outcomes across the continuum of care [26].

There was a significant association between viral load suppression and participants' HCW preference for receiving hospital service from a familiar and local HCW. When participants were asked whether they cared whether or not the HCW who cared for them was a familiar person from their neighborhood, those who were less picky about who cared for them had higher rates of viral suppression [27–29].

Even though 61.2% of PLHIV prefer to keep their HIV status a secret, 93.1% of PLHIV confirmed disclosing their health status to at least one person around them. Additionally, this encouraged them to take medication (93.6%) and even be supported financially and socially (29.5%). In our study, disclosing HIV status did not have a significant association

with viral load suppression. However, the literature suggests that the disclosure of HIV status increases the chances of virological suppression by improving adherence to ART, as was observed in Uganda [25].

This study has some limitations, including the cross-sectional design, which limits the ability to establish causal relationships. Additionally, the study participants were recruited from two regional referral hospitals in Tanzania, which may limit the generalizability of the findings to other settings in Tanzania. Moreover, the study relied on participants' accounts of how they were treated differently because of their status. Self-reported data are frequently subject to recall bias and social desirability. To reduce this bias, some information, such as viral load information, was obtained from hospital records. Additionally, patients were requested to skip questions they did not remember rather than guessing the responses.

## 5. Conclusions

Self-perceived stigma seemed to affect how well PLHIV in Dar es Salaam, Tanzania, could suppress their viral load. HIV patients in the Dar es Salaam area had significantly lower viral loads when gender and time between refills were considered.

This study highlights the need for strategies that support HIV status disclosure, address self-discrimination and stigmatization, and promote ART adherence among people living with HIV. Such strategies should include counseling on stigmatization and its effects on PLHIV, the use of reminder tools, and the creation of safe spaces for people living with HIV to access quality healthcare services without fear of discrimination and stigmatization. Future studies could explore the effectiveness of such strategies in improving the health outcomes of people living with HIV.

**Author Contributions:** Conceptualization, M.S.K. and R.Z.S.; methodology, M.S.K., I.H.M., R.Z.S. and G.L.S.; validation, G.M.B., R.Z.S. and G.L.S.; formal analysis, M.S.K. and R.Z.S.; writing—original draft preparation, M.S.K. and R.Z.S.; writing—review and editing, M.S.K., I.H.M., G.M.B., G.L.S., R.Z.S. and J.K.; visualization, M.S.K. and R.Z.S.; supervision, J.K. All authors have read and agreed to the published version of the manuscript.

**Funding:** This research received no external funding.

**Institutional Review Board Statement:** Ethical clearance was sought from the MUHAS Institutional Review Board (DA.25/111/01/10/02/2021). Temeke RRH and Amana RRH permitted the study to be conducted on their respective premises and with patients. All participants provided written informed consent for participation. The consent included information on the study description, data privacy/confidentiality, and handling. Patients who reported self-discrimination and stigmatization were counselled on medication adherence. Preventive measures for the COVID-19 pandemic were observed during the data collection period—centers for Treatment and Counseling for their dedicated cooperation during the data collection phase.

**Informed Consent Statement:** Informed consent was obtained from all subjects involved in the study.

**Data Availability Statement:** Data can be made available upon request.

**Conflicts of Interest:** The authors declare no conflict of interest.

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
