# Peer review of "Factors Associated with Viral Load Suppression and Indicators of Stigma among People Living with HIV in Dar es Salaam Tertiary Hospitals, Tanzania"

_2036-7481, doi:10.3390/microbiolres14020050_

Round 1
Reviewer 1 Report
Review comments
The authors analyzed small population in Tanzanian urban area to identify factors that are associated with HIV viral load suppression. PLWHIV were questioned at two different hospitals and the study relied on self-based and self-reflected answers. Statistical analysis was conducted to determine the significance of socio-demographic and other parameters that correlate with viral load suppression. The article raised important questions of self-stigmatization and several factors need to be incorporated to significantly improve the manuscript.
Major comments:
1. Study population and sampling. It is unclear whether the PLWHIV were tested for viral load or the values were obtained from the clinical records. The study design indicated that between July and August participants were sampled in “n” intervals. It is important to specify whether participants were tested by the study team, or whether it was self-reported from the most resent visit or taken from clinic records. Need to specify in the methods and results.
2. Table 3 is inconsistent with the number of study participants. Please correct the number of participants that answered each question as listed in tables 1, 2 and 4. For example: the first question in tables 3 “preferred CTC location (N=399)”, however when add all answers N=406. The second question: “Why prefer separated HIV CTC” the number of answering participants is listed, however when add all answers together n=471. Every question in table 3 has more answers than participants. The authors should indicate if the participants gave several answers and, in case of the mistake, should correct their numbers for each question in table 3.
3. Factors associated with viral load suppression. The authors state the “Participants with a refill interval of three months had more viral suppression than those with six months and additional months”. Tables 4 and 5 contradict to that statement. From the data and statistical analysis, I can tell that the most viral suppression was observed in those with 6 months refill. In table 4 only 4% of participants that refill every six months had high VL while 16.3% participants had high VL among those refill every 3 months. Did they use absolute number? If we look at the percentage the six months refill regiment demonstrated better viral suppression.
4. In table 5 why use six months as a reference? What is the primary covariate used for the multivariate logistic regression model? What are the factors the model is adjusted for? This needs to be written in the methods as a formula that was used and described in the results.
Minor comments:
Table 2: disclosure of HIV status: “the felt stigmatized me”… do the participants mean that they were stigmatized after disclosing their HIV serostatus? If yes, please change to “the left me stigmatized”. If the participants mean another feeling, please translate accordingly.
Section 3.4. Factors associated with viral load suppression. The sentence: “Participants with a refill interval of three months had 5 more viral suppression than those with six months and additional months of refill interval, with a p-value of 0.000 (p-value = 0.004) (Table 4)” the p value 0.000 is associated with Table 4, while the p value 0.004 is coming from table 5. This needs to be referenced appropriately.
Table 2: disclosure of HIV status: “the felt stigmatized me”… do the participants mean that they were stigmatized after disclosing their HIV serostatus? If yes, please change to “the left me stigmatized”. If the participants mean another feeling, please translate accordingly.
Reviewer 2 Report
I appreciate the opportunity to critically review the manuscript titled "Factors associated with viral load suppression and indicators of stigma among people living with HIV in Dar es Salaam tertiary hospitals, Tanzania" in which the authors assessed how stigmatization and self-stigmatization affect viral load in adults living with HIV.
Comment 1: It would be interesting for readers to know the total population (stratified by sex) of patients receiving antiretroviral therapy in the study units during the recruitment period. It would also be of interest to know, in general terms, the age and sex structure of the subjects who were not recruited.
Comment 2: I am struck by the fact that almost three-quarters of the participants were women. In the vast majority of populations, the HIV epidemic is heavily skewed towards young men. Therefore, how representative is the analyzed sample of the specific population of the analyzed locations? Is there a possibility of selection bias? It would be interesting to briefly discuss this fact.
Comment 3: I believe that section 3.4 and Table 4 should ideally clearly specify the proportion of participants classified in the viral load suppression group. If this proportion is greater than 10%, it would be important to discuss the effect of a frequent event on the obtained odds ratios.
Comment 4: Overall, I believe that the Discussion should be substantially enriched. There are multiple aspects of the results that would be worth discussing for the benefit of the article's quality.
Comment 5: I am concerned about the finding related to the effect of the patient's gender on viral suppression, especially because it contradicts previous publications. This is strongly related to the comment 2 made by me. Is there a possibility of selection bias in the study sample? I suggest discussing this in detail.
I would like to congratulate the authors on their interesting work on a topic of high clinical and epidemiological relevance. However, I believe that the final product can be enriched.
Round 2
Reviewer 1 Report
The authors addresses my questions and adjusted the manuscript accordingly.
Reviewer 2 Report
I appreciate the opportunity to review once again the manuscript entitled "Factors associated with viral load suppression and indicators of stigma among people living with HIV in Dar es Salaam tertiary hospitals, Tanzania." I reiterate the significance of the topic and congratulate the authors on their approach, as well as for addressing the comments provided by myself.
I believe that the quality of the manuscript has significantly increased and it will be of great interest to the readers of the Journal.